# Low-Power Scan Correlation-Aware Scan Cluster Reordering for Wireless Sensor Networks

**DOI:** 10.3390/s21186111

**Published:** 2021-09-12

**Authors:** Sangjun Lee, Kyunghwan Cho, Jihye Kim, Jongho Park, Inhwan Lee, Sungho Kang

**Affiliations:** Electrical and Electronic Engineering Department, Yonsei University, Seoul 03722, Korea; lsj920807@soc.yonsei.ac.kr (S.L.); herrcho79@soc.yonsei.ac.kr (K.C.); irene152@soc.yonsei.ac.kr (J.K.); jongho0117@soc.yonsei.ac.kr (J.P.); hwan2996@soc.yonsei.ac.kr (I.L.)

**Keywords:** wireless sensor networks, cryptography, design for testability (DFT), low-power testing, scan chain reordering

## Abstract

Cryptographic circuits generally are used for applications of wireless sensor networks to ensure security and must be tested in a manufacturing process to guarantee their quality. Therefore, a scan architecture is widely used for testing the circuits in the manufacturing test to improve testability. However, during scan testing, test-power consumption becomes more serious as the number of transistors and the complexity of chips increase. Hence, the scan chain reordering method is widely applied in a low-power architecture because of its ability to achieve high power reduction with a simple architecture. However, achieving a significant power reduction without excessive computational time remains challenging. In this paper, a novel scan correlation-aware scan cluster reordering is proposed to solve this problem. The proposed method uses a new scan correlation-aware clustering in order to place highly correlated scan cells adjacent to each other. The experimental results demonstrate that the proposed method achieves a significant power reduction with a relatively fast computational time compared with previous methods. Therefore, by improving the reliability of cryptography circuits in wireless sensor networks (WSNs) through significant test-power reduction, the proposed method can ensure the security and integrity of information in WSNs.

## 1. Introduction

Wireless sensor networks (WSN) are networks in which data obtained by observing the environment by a large number of sensors deployed in a specific area are sent securely to other sensors or centers in the network. These networks have the following capabilities: not connected to a central node, self-managing and healing, not connected to a specific network topology, multiway routing, preserving the integrity and confidentiality of data, and robustness [1]. For these reasons, WSNs have been widely used for applications in environmental, health, military, and commercial systems, such as intelligent transportation, smart home, industrial monitoring, logistics, health care, among others [2,3]. However, WSNs have serious problems in privacy and security [4,5]. In addition, WSNs have some dependability problems because WSNs are increasingly used in critical application scenarios where the level of trust on WSNs becomes an important factor, affecting the success of large-scale industrial WSN applications [6]. To solve the problems, cryptography is performed to ensure the security and integrity of information in WSNs. These cryptography algorithms require extensive modular multiplication and exponentiation operations, which are much more computationally expensive. Therefore, to obtain high computational power, the circuits for cryptographic algorithms are required. For the accuracy of the cryptographic algorithm, cryptographic circuits must be rigorously tested to guarantee their quality [7]. Therefore, a scan architecture known as design for testability (DFT) technique, which can obtain high test coverage, is generally used to test the cryptographic circuits [8]. However, the cryptography circuits consume a high power because they require high computational power. In addition, the power consumption during scan testing is much higher than that in functional operation. According to a study report, the average test power can be three times the power consumed during a functional operation [9]. Moreover, power consumption during scan testing becomes a more serious problem as the number of transistors and the complexity of chips is increased. Excessive power consumption reduces circuit reliability and results in a yield loss caused by IR drop; therefore, the test power consumption must be minimized.

In this paper, to solve these problems, a novel scan correlation-aware scan cluster reordering method is proposed. The novelty of the proposed method is that using a new scan correlation-aware clustering in order to place highly correlated scan cells adjacent to each other with a small computation time, the proposed method has a significant power reduction effect compared to that of the previous methods. The performance of the proposed method is compared with state-of-the-art methods with experiments on the International Symposium on Circuits and Systems (ISCAS)’89 and OpenRISC (OR)1200 benchmark circuits. The experimental results demonstrated that the total number of shift transitions was reduced by 28.31% on average compared with the pervious methods with minimal computational time, confirming the effectiveness and superiority of the proposed method for shift-power consumption and computational time. Therefore, by improving the reliability of cryptography circuits in WSNs through significant test-power reduction, the proposed method can ensure the security and integrity of information in WSNs and contributes to applications in environmental, health, military, and commercial systems, such as intelligent transportation, smart home, industrial monitoring, logistics, and health care, where WSNs are used.

The remainder of this paper is organized as follows. Section 2 describes the related works on scan-based testing for low-power testing. The hierarchical agglomerative clustering (HAC) technology and the weighted transition metric (WTM), a background for the proposed method, are introduced in Section 3. The portion of X-bits in the test patterns and the potential of using X-bits for power consumption are also discussed. Section 4 discusses the correlation distribution of adjacent scan cells, a motivation for the proposed method. The central concept of the proposed method and a step-by-step example are also presented. In Section 5, the experimental results of the power consumption and computational time are discussed. Finally, Section 6 concludes this paper.

## 2. Related Works

Generally, the test-power consumption problem caused by excessive switching has been solved by the following three methods: (1) automatic test pattern generation (ATPG), (2) hardware modification, and (3) scan chain reordering.

ATPG-based techniques control the configurations of the test pattern generation. Minimum transition (MT)-filling uses X-bits (don’t care bits) in the test patterns to reduce test power [10]. The concept is to fill X-bits with care bits to lower the shift-in transitions. Li et al. proposed a new X-filling technique, called “iFill,” that considers the impact of X-bits on the switching activities of the circuit nodes and reduces power consumption during both scan-in and -out operations [11]. Devanathan et al. proposed a glitch-aware pattern generation and optimization framework for a power-safe scan test [12]. Bosio et al. proposed a power-aware test pattern generation method for at-speed launch-off-shift (LOS) testing [13]. Furthermore, Girard et al. proposed a test vector ordering technique for switching activity reduction during scan testing [14]. Sivanandam et al. proposed a new power transition x-filling based selective Huffman encoding technique, which achieves test data compression and switching power reduction [15]. These methods reduce the test power consumption without requiring any design modification or additional DFT circuitry. However, applying them to an actual test flow is difficult because their power reduction is less effective than the hardware-based or scan chain reordering methods.

The hardware-based solution adds extra DFT circuitry or modifies the design. Chiu et al. proposed a jump scan technology that shifts two bits of scan data per clock cycle such that the scan clock frequency is halved without increasing the test time [16]. Bonhomme et al. proposed a gated clock scheme for the scan path and a clock tree to feed the scan path [17]. Rosinger et al. proposed a scan architecture with mutually exclusive scan segment activation for shift- and capture-power reduction [18]. Zhang et al. presented a multiphase clock scan technique for low test power [19]. Furthermore, Lee et al. proposed a built-in self-test scheme that uses a new scan partitioning technique and a decoding methodology for low test power [20]. Cao et al. proposed a bypassable scan data retention flip-flop (BPS-DRFF), which prevents the signals from going to the combinational circuit during scan shifting, for low-power testing [21]. These methods reduce the test power during scan testing but are likely to cause area overhead or performance degradation and serious delays [22].

Scan chain reordering is a solution that changes the order of scan cells to minimize shift transitions. Although the scan chain reordering method has routing overhead, it is widely applied to reduce test power because of its high performance and simple architecture. Accordingly, Cui et al. described a k-means clustering-based scan chain reordering method under a routing constraint [23]. Seo et al. presented a statistic-based scan chain reordering method that performs scan partitioning to reduce routing overhead [24]. Huang et al. described a routing constrained scan chain reordering technique [25]. These methods determine the optimal scan chain reordering sequence that considers both routing overhead and low-test power. However, these methods can be an overhead of the overall design flow, which are performed using the layout information in a back-end design flow and executes the layout twice to reduce the test power simultaneously under the routing constraint. In addition, these methods do not effectively reduce the test power due to using both routing and test power information.

The method in [26] used both response and pattern correlations to minimize the scan-out and -in transitions simultaneously. Seo et al. proposed a scan chain reordering-aware X-filling and stitching method for the scan-shift power reduction [27]. Pathak et al. proposed a logic cluster controllability (LoCCo)-based scan chain stitching method that reduces the shift-in power [28]. The method in [29] merged the scan cells with higher care bit density toward the front of scan chains. Kim et al. proposed a new shift mechanism of exclusive scan-shift and the scan chain reordering method considering the number of the scan cell’s fan-outs [30]. Lee et al. proposed the scan chain stitching method using the circuit topology to analyze the testability of each flip-flop [31]. However, these methods are not effective in reducing power, and the computational time is also not negligible. Therefore, a scan chain reordering method with a significant reduction in power and a fast computational time must be developed.

In this paper, a novel scan correlation-aware scan cluster reordering method is proposed to significantly reduce the test-power consumption with a small computation time. In order to achieve this, the algorithmic innovation is derived as follows. The proposed method consists of three steps. First, a pairwise distance matrix between scan cells is constructed using test patterns. A weighted distance matrix must be prepared in advance to consider both shift-in and -out transitions. Second, scan correlation clustering is performed using the distance matrix built in the previous step, and a resultant dendrogram tree is obtained after the clustering. The sequence of the leaves of the dendrogram tree, which are the newly aligned scan cells, is used in the third step to reorder the scan chain. Third, scan chain reordering is performed using the sequence of the newly aligned scan cells.

## 3. Background

This section introduces the HAC technology incorporated into the proposed method. The portion of X-bits in the test patterns and the potential to use the X-bits is then discussed. Finally, a WTM widely used to predict the scan-shift power is introduced.

### 3.1. Hierarchical Agglomerative Clustering (HAC) Technology

HAC technology is widely used as a method for classifying objects in unsupervised machine learning and has multiple applications, such as pattern recognition, computational biology, and data mining [32]. The clustering method builds a solution by initially assigning each object to its own cluster and then repeatedly selecting and merging cluster pairs to obtain a single all-inclusive cluster. Hence, it builds a tree from the bottom, referred to as a dendrogram tree. The main parameters in the HAC method are (1) the metrics used to compute the similarity of objects and (2) the method used to determine the cluster pair to be merged at each step. This study attempts to solve the scan chain reordering problem for low test power using scan correlation information and HAC technology. In the proposed method, three steps are required to combine scan correlation information and HAC technology. Section 4 elaborates on the three steps of the proposed method and the two main parameters to reduce both shift-in and -out transitions.

### 3.2. Portion of Don’t Care Bits in Test Patterns

Care bits are bits in the test patterns corresponding to the scan cells that have been assigned binary values during test pattern generation. X-bits are bits in the test patterns other than the care bits. A considerable number of the total bits in test patterns are X-bits [10]. Figure 1 is an example of the X-bit distribution of the s38584 benchmark circuit. The first few ATPG patterns have a large number of care bits, but the number of care bits rapidly decreases as the pattern generation proceeds. These X-bits can be exploited to reduce power consumption during scan-based testing. The MT-filling technique, in which an X-bit is set to the last care bit, is a widely preferred method for reducing shift-in transitions. For a circuit with many X-bit portions in the test patterns, the probability of finding a scan cell pair with a strong positive correlation increases if the X-bits are filled with specified bits. If we reorder the scan chain such that these highly correlated scan cells are adjacent to each other, the total number of shift transitions can be reduced, which indicates that the power consumption can be lowered.

### 3.3. Weighted Transition Metric (WTM)

A WTM, which was proposed in a previous study [10], is widely used as a method for estimating the scan-shift power and defined as follows:(1)$$WTM_{i}=\overset{N-1}{\underset{j=1}{\sum }}(S_{i,j}⊕S_{i,j+1})\times j$$

In Equation (1), $S_{i,j}$ is the logic value of the jth scan cell in the ith test vector. The number of shift transitions depends not only on the bit difference between the adjacent scan cells but also on the position of the bit difference in the scan chain. Equation (1) calculates the number of shift transitions, reflecting these two factors. The smaller the WTM, the lower the scan-shift power. Therefore, this study aims to reduce the WTM by increasing the correlation of the adjacent scan cells and reordering the scan chain.

## 4. Proposed Method

This section first presents the characteristics of the correlation distribution between the adjacent scan cells, which is the motivation of the proposed method. The relationship between the adjacent scan correlation and the power consumption and the possibility of increasing the correlation are also discussed. The details of the proposed method are then described. The proposed algorithm consists of three steps, which are described, including example test patterns. Moreover, the effectiveness of the proposed method is confirmed by comparing the WTMs before and after the scan chain reordering.

### 4.1. Motivation

Consider the ISCAS benchmark s38584 circuit to illustrate the concept of correlation distribution. Figure 2 illustrates the correlation distribution of the adjacent scan cells when stitched in alphabetical order. This correlation distribution can be obtained as follows. First, a scan chain is stitched alphabetically (a conventional method), and the ATPG is run to obtain the test patterns. Second, the correlation between the adjacent scan cells is obtained by calculating the normalized hamming distance and subtracting it from 1. This procedure is repeated for all adjacent scan cells to obtain a correlation distribution of a circuit. Figure 2 illustrates that the correlation distribution is densest near 0.9 and widely distributed with a wide tail stretched far to the left in a correlation area between 0 and 0.9.

The correlation value is an indicator of the degree of similarity between the scan cells. The lower the correlation between the scan cells, the more likely the opposite phase is in the test patterns. A low correlation indicates that switching activities occur mostly between the scan cells during the scan-shift operation. Therefore, a high correlation between adjacent scan cells is advantageous for shift power.

Figure 3 illustrates the heatmap graph of all scan cell-to-cell correlations in four ISCAS benchmark circuits. The closer the color is to green, the higher the correlation. In contrast, the closer the color is to red, the lower the correlation. The scan cells of a weak correlation (red grid, Figure 3) are expected to cause switching activities to occur excessively during scan testing. Therefore, scan cells with a high correlation must be found and placed adjacent to each other to reduce the switching activities. The proposed method applies this approach using a novel scan correlation clustering algorithm. The technique merges high-correlation scan cells while gradually clustering them using the correlation information between the scan cells. The correlation information is obtained from the test patterns, which accurately indicate the switching activities between the scan cells. Consequently, the proposed method can reduce the shift power.

### 4.2. Overall Flow

The goal of the proposed method reduces the test-power consumption during scan testing in cryptographic circuits and improve the reliability of cryptographic circuits to guarantee their quality in the manufacturing test. To do this, the proposed method based on the scan chain reordering is performed because of its ability to achieve high power reduction with a simple architecture and can apply to the design circuits with the scan architecture. Therefore, through the proposed method, the scan cells with high correlation are located adjacent to each other to obtain a significant power reduction effect, and, at the same time, reduces the burden on the design because additional hardware is not required.

To perform the proposed method, the test patterns are required as input data, and these test patterns can be easily obtained by using EDA tools. First, the tools synthesize the benchmark circuits and insert scan architectures. Next, the test patterns are obtained after running the ATPG to detect the targeted faults and contain a binary value for each scan cell during the scan-in and -out operations. The input vector is used during the scan-in operation while the response vector is used during the scan-out operation. The switching operation is caused by a logic difference between the adjacent scan cells during the scan-in and -out operations. Therefore, the analysis of the input and response vector reveals each scan cell’s switching behavior during the scan-in and -out operations. The opposite phases between the adjacent scan cells cause switching activities during scan testing; hence, scan cells with a similar phase must be grouped. This grouping information can be obtained by using the correlation distribution between all scan cells. When reordering scan cells based on the correlation distribution, the switching activities will be reduced.

The proposed idea is to group scan cells of high similarity by analyzing the test patterns and place the scan cells adjacently when stitched into scan chains. The proposed method consists of three steps, as shown in Figure 4. First, the distance matrix of the scan cells is constructed by calculating the pairwise distance of the scan cells using the test patterns. Second, scan correlation clustering is performed using the distance matrix, and the resultant dendrogram tree is obtained. Third, the scan chain is stitched in the order of the first-to-last item of the dendrogram leaves. The scan cells with a high correlation are then stitched sequentially, reducing the total number of scan-shift transitions. After this scan chain reordering, the mean of the adjacent scan correlation distribution is shifted to a high value, and the standard deviation is moved to a low value (depicted experimentally in Section 5). Additionally, these three steps can be implemented by an automated system such as the EDA tool. The overall flow of the proposed method is explained for each step with an example test pattern.

### 4.3. Construction of Distance Matrix

If a fault occurs near the scan-in cell, the test stimulus will contain more errors than when a fault occurs near the scan-out cell. Therefore, the failure in the test response would have spread more than in the case where it is close to the scan-out. Figure 5 illustrates an algorithm for constructing the distance matrix of the scan cells. Both the input and response vectors must be prepared to analyze the correlation between the scan cells during the scan-in and -out operations. The proposed method focuses on reducing shift-power consumption, but the capture power is also critical in digital circuits. Therefore, the 0-filling methodology is applied when generating the test patterns, which is known to reduce the peak capture activity [33]. However, although 0-filling is used to reduce both capture and shift power, the proposed method can use the test patterns of the various filling methodologies. The objective of this step is to construct the distance matrix between all scan cells, which is required for the scan correlation clustering in Step 2. The distance between the scan cells must be defined before constructing the distance matrix. The test patterns contain a logic value of each scan cell; thus, it is reasonable to define the distance between the scan cells as a normalized Hamming distance. Consider two scan cells, $FF_{i}\;\mathrm{and}\;FF_{j}$, which are two randomly chosen flip-flops; $h_{ij}$ is the normalized hamming distance between the two scan cells and defined as follows:(2)$$h_{ij}=H(FF_{i},FF_{j})/k$$

In Equation (2), H(i,j) denotes the number of different bits in the test patterns between the $FF_{i}\;\mathrm{and}\;FF_{j}$ scan cells, and *k* denotes the number of patterns. As shown in lines 5 to 10 in Figure 5, the distance matrix is constructed by calculating $h_{ij}$ between all the scan cell pair combinations.

The goal of the proposed method is to reduce the total shift transitions; both the input and response vectors’ Hamming distance matrix must be weighted and summed to build the total Hamming distance matrix, as shown in Figure 5. If shift-out transitions are higher than shift-in transitions, adding more weight to the response_dist_matrix, which includes the correlation information of the scan cells in the response vector, is advantageous for reducing the total number of shift transitions. However, the number of shift transitions cannot be obtained before the proposed method is applied. Therefore, it is required to determine the weight in advance, which correlates with the number of shift transitions. In the proposed method, the relative standard deviation (*RSD*), a standard deviation/mean of the adjacent scan correlation distribution, is used to determine the weight. If *RSD* is high, that means the number of shift transitions is also high because the correlation of the adjacent scan cells is low and widely distributed. Conversely, if *RSD* is low, that means the number of shift transitions is also low because the correlation of the adjacent scan cells is high and narrowly distributed.

The experiments also demonstrate that *RSD* of the adjacent scan correlation distribution is proportional to the number of shift transitions shown in Figure 6 (discussed in more detail in Section 5). Therefore, to determine the weight of the Hamming distance matrix, the proposed method calculates the input and response vectors’ *RSDs* of the adjacent scan correlation distribution and uses these values as *input_weight* and *output_weight*. Consequently, the *RSD* of the adjacent scan correlation distribution can be used as a weighting factor. Finally, the total Hamming distance matrix is created and then used in the next step.

Consider the example of reducing the shift-out transitions using the proposed method. The shift-in transitions are not considered in this example, to simplify the explanation. First, a response vector is prepared by performing scan chain stitching with the conventional method, where scan chains are stitched alphabetically, and by running the ATPG with a 0-fill. Figure 7a illustrates the example response vector, which has seven scan cells and ten vectors.

The total number of shift-out transitions is 111 based on the WTM, and five out of the six distance values of the adjacent scan cells are greater than or equal to 0.5. Second, a scan cell-to-cell distance matrix is created by calculating the Hamming distance between each scan cell pair, as shown in Figure 7b. For example, the distance between FF0 and FF1 is 0.5 because 5 out of the 10 binary values in the response vector are different. Based on the distance matrix of Figure 7b, 9 out of the 21 scan cell pairs have distance values less than or equal to 0.4. Therefore, the correlation between the adjacent scan cells can be increased through scan chain reordering by using a scan cell pair with low distance (high correlation).

### 4.4. Scan Correlation Clustering

Figure 8 illustrates the scan correlation clustering algorithm. In Figure 8, step 2 begins with each scan cell in a separate cluster. During each procedure, the two most-similar scan cell clusters are joined into a single new scan cell cluster. The intercluster distance must be defined to find the two most-similar scan cell clusters because the cluster pairs with the closest intercluster distance can be considered most similar and should be merged.

The average intercluster distance metric is used to evenly consider all the scan cells in a cluster and defined as follows:(3)$$dist(C_{h},C_{r})=\frac{1}{n_{i}n_{j}}\sum_{FF_{i}\in C_{h},\;FF_{j}\in C_{r}}h_{ij}$$

Equation (3), which is used in the second step of Figure 8, adds all the distances of the scan cell pairs between the clusters and divides by the number of scan cell pairs. The previously obtained $h_{ij}$ that represents the Hamming distance between $FF_{i}$ and $FF_{j}$ is used to calculate the distance between the scan cell pairs. After selecting and merging pairs of clusters with the minimum distance, the distance matrix must be updated for a new cluster. Moreover, the cluster pairs are never separated once they are combined in each clustering step. This procedure iterates until only one cluster is left. The total number of iterations performing the scan correlation clustering is the total number of scan cells minus 1 because two small clusters are merged at a time. Finally, the dendrogram tree can be obtained. The dendrogram tree represents the successive agglomerations, starting from one scan cell per cluster until all scan cells belong to one cluster. The leaves at the bottom correspond to a scan cell and are used to decide in which order to stitch. The *y*-axis of the dendrogram tree denotes the distance of the scan cell cluster pairs, where a larger value indicates a higher level of dissimilarity. Figure 9a–f illustrates the simple example of a dendrogram tree of the scan correlation clustering process for each step. It begins with each scan cell being assigned to its own cluster. The clustering is performed six times because the clustering process proceeds by repeatedly combining the two small clusters, with a total of seven scan cells. The red line represents two clusters merged at each step. The cluster pair with the shortest distance is found from the distance matrix obtained from the previous step.

The distance of cluster FF1–FF3 is the shortest at 0.1; hence, the FF1–FF3 cluster is merged into one cluster, as shown in Figure 9a. The distance matrix is then updated by recalculating the distance between the FF1–FF3 cluster and the other clusters. In the next step (Figure 9b), FF1–FF3–FF6 becomes one cluster because the distance of the FF1–FF3 and FF6 clusters is shortest at 0.15. This process is iteratively performed until the last all-inclusive cluster remains, resulting in the rightmost dendrogram tree, as shown in Figure 9f. The scan cell list (FF1, FF3, FF6, FF4, FF0, FF2, FF5), which corresponds to the leaves of the dendrogram tree, is then obtained.

### 4.5. Scan Cluster Reordering

The scan cluster reordering is a final step of the proposed method and determines the order of scan cells. Based on the dendrogram leaves in Figure 9f, the high-correlation scan cells are placed adjacent to each other after the scan correlation clustering. Therefore, the correlation between the adjacent scan cells can be assumed to increase if scan chain stitching is performed in one direction of the leaves. In Step 3, scan chain stitching is performed in the order of the first-to-last item of the dendrogram leaves for low test power. Consider the example response vector after applying the proposed method in Figure 10. The distance between the adjacent scan cells is less than or equal to 0.5, which is much improved compared with the conventional scan chain stitching in Figure 7a. When considering the FF4 and FF5 scan cell pair, the distance between those scan cells is 0.9; moreover, they were adjacent before the scan chain reordering, which indicates heavy switching behavior. However, they are relocated to be adjacent to the other scan cell with a similar phase after applying the proposed method. Consequently, the total number of shift-out transitions are reduced from 111 to 48, a 57% reduction when compared with the conventional stitching method.

The proposed method can achieve a significant power reduction effect during the scan testing by reordering the highly correlated scan cells to be adjacent. However, when changing the order of scan cells significantly, the routing length increases compared with that of the conventional stitching method based on the physical design—resulting in performance overhead (e.g., latency). Therefore, it is important to decrease the routing length when the proposed method is applied. In the conventional DFT setup, the placement-aware stitching is generally used to reduce the total routing length. The scan partitioning also is performed to relieve the routing overhead between the scan cells, limiting the range of the scan chain stitching by grouping the near scan cells. In the proposed method, this scan partitioning information is used to minimize the routing length. First, the place-aware stitching and scan partitioning are performed by considering the physical layout similar to the conventional DFT setup, and the information of the scan partitions was extracted. The proposed method limits the range of the scan chain reordering to only the scan cells within the scan partitions where each scan partition has a single scan chain. The proposed method can apply the scan chain reordering within each scan partition since the distance matrix can be calculated according to the scan length in a scan chain.

### 4.6. Test Compression Mode

The proposed method only changes the order of scan cells to minimize test-power consumption and can be scalable to the design methodologies related to the scan architecture. Recently, a test compression is widely used to compress the test data, and it is important to manage the test compression architecture. For the designs including the test compression architecture, the proposed method of managing test compression follows. First, the conventional test compression in alphabetical order is performed on the target design, and the original test patterns are generated using the conventional test compression. By using these test patterns, the distance matrices are created to obtain correlation distribution between the scan cells. In addition, to maintain the number of scan chains, the proposed method creates the distance matrices to be as many as the number of scan chains. Next, hierarchical agglomerative clustering (HAC) technology is performed for the distance matrices and the dendrogram trees are obtained. Finally, the order lists of the scan cells for the low-power testing are obtained through the dendrogram trees, and the new test compression is performed by reordering the scan cells. At this time, not only the order of the scan cells is changed, but also the test compression architecture (i.e., de-compressor and compressor logic) is re-created according to the order of scan cells to be changed. In addition, the new test patterns are generated due to the change of the test compression architecture.

## 5. Experimental Results

### 5.1. Evaluation Methodology

In the experiments, the performance of the proposed method was evaluated on the ISCAS’89 and OpenCores benchmark circuits to compare the shift-power reductions. The total number of shift transitions were evaluated using the WTM, which is a widely employed estimation method for shift power, as discussed in Section 3.3. Synopsys Design Compiler was used to synthesize the benchmark circuits, Synopsys DFT Compiler was used for the DFT insertion, and Synopsys TetraMax was used to generate patterns and apply X-filling technology. The proposed method was compared with the pattern-based scan chain reordering method [26,27] and the logic topology-based scan chain stitching method [28] for shift-power reduction. The experiments were conducted using the SAED 32 nm library supported by the Synopsys ARMENIA Education Department.

### 5.2. Correlation Distribution

For confirming the effectiveness of the proposed method, the correlation distributions of the adjacent scan cells were evaluated with both the proposed and conventional methods, in which the scan chains were stitched alphabetically with the adjacent fill technology applied. The experiments were conducted on four ISCAS’89 circuits and one OpenCores circuit using a kernel density estimation (KDE) plot. KDE is a method of estimating the probability distribution density using a kernel function, such as a Gaussian distribution. A KDE plot is beneficial for simultaneously visualizing multiple distributions compared with a histogram because it smoothes the histogram such that the change in the distribution is visually comprehensible.

As shown in Figure 11, the number on the *x*-axis denotes the correlation value of the adjacent scan cells, while the number on the *y*-axis indicates the probability distribution density. The blue region represents the proposed method, while the red region illustrates the conventional stitching method. The higher the *y* value, the more the correlation of the adjacent scan cells is distributed in the corresponding *x*-axis region. After applying the proposed method, the mean value of the proposed method was increased while the standard deviation was decreased. Therefore, the *RSD* (standard deviation/mean) of the proposed method was decreased. The reduced *RSD* of the correlation distribution indicates that the correlation distribution is narrowly concentrated at high values, which is advantageous for shift-power reduction. Therefore, the correlation distribution of the proposed method was improved such that the switching activities between the adjacent scan cells were reduced for all experimental benchmark circuits.

Table 1 presents the comparison of the correlations of the adjacent scan cells and the power consumption with the conventional stitching method. The WTM was reduced by 58.28% on average without affecting the test coverage. The *RSD* of the adjacent scan cells’ correlation was also reduced by 60.31%. Moreover, the *RSD* of the adjacent scan cells’ correlation has a linear relationship with WTM, which provides an experimental basis for weight calculation of the distance matrix.

### 5.3. Power Consumption during Scan-Shift Operation

In the experiments, the WTMs were measured to confirm the superiority of the proposed method for shift-power reduction compared with the previous method. The performance of the proposed method was compared with state-of-the-art methods [26,27,28] using four ISCAS’89 circuits, as shown in Table 2. The proposed method outperformed that in [28], which is one of the latest scan reordering methods, by an average of 43.46%. In [28], the care bits in test vectors are consolidated toward the beginning of the scan chains, and don’t care bits are located toward the end of the scan chains. Hence, the method in [28] has the advantage of reducing the shift-in transitions compared with the conventional method. However, it is not effective in reducing the total number of shift transitions because the shift-out transitions are not considered. Shift-in transitions can be reduced through X-filling technology; however, the response from the combinational logic is random. Consequently, the shift-out transitions tend to be much larger than the shift-in transitions. Therefore, the shift-out transitions must be considered. The proposed method also outperformed the methods in [26] and [27], which considers both the shift-in and -out transitions. The experimental results demonstrated that when applying the proposed method, the power consumption was decreased by 28.31% on average in all the experimental benchmark circuits compared with the previous method. The experimental results demonstrate that the proposed method is effective in scan chain reordering for low test power.

In addition, for a circuit with many X-bit portions in the test patterns, the probability of finding a scan cell pair with a strong positive correlation is increased. The experiments were conducted on four ISCAS’89 circuits and one OpenCores circuit to investigate the power reduction effect of the proposed method based on the X-bit portion. The WTMs of the proposed and conventional methods were compared using circuits with various X-bit portions and X-filling (0, adjacent) methodology. The improvement was then evaluated by handling the WTMs of shift-in and shift-out separately. As shown in Table 3, the proposed method has a significant power reduction effect compared to that of the conventional methods with 0-fill and adjacent-fill. Particularly, it can be seen that the shift-out transitions of the proposed method are reduced greatly. This is because the proposed method effectively places highly correlated scan cells adjacent to each other. For or1200 circuit with a 99% X-bit portion, the power reduction rate was 70.80%, which is the highest among the benchmark circuits. Touba [34] asserted that the X-bit density of industrial circuits was 95–99%. Even if the test patterns have low X-bit density, the proposed method can also effectively reduce the power consumption, as shown in Table 3. Therefore, the proposed method can achieve a power reduction effect for the test power in industrial circuits compared with the conventional method.

### 5.4. Computational Time

The proposed method consists of three steps. Step 3 has a negligible effect on the computational time because no computational process is involved in this step. In contrast, Steps 1 and 2 might cause excessive computational time because, as the circuit size increases, the number of scan cells also increases; the computational time can be excessive due to the iterative computational processes involved in these steps. In the experiments, the computational time of the proposed method was compared with that in [28] using four ISCAS’89 circuits and one OpenCores circuit on the system with the following hardware specifications: Intel(R) Core (TM) i5–4690 CPU at 3.50 GHz with 8 GB RAM.

As presented in Table 4, the computational time is significantly reduced by 91.72% on average compared with that in [28]. In the computational time of the proposed method, based on the or1200 circuit, the computational time is significantly increased compared with the rate at which the number of scan cells increased. However, for large industrial circuits with a substantial number of scan cells, multiple scan chains are used. In general, when the number of scan cells is increased, the number of scan chains is also increased to maintain the proper scan length. Therefore, the computational time for a distance matrix is almost the same for the same scan length.

The only difference is the number of distance matrices. Let *S* be the number of scan cells and *M* be the number of scan chains, respectively. Then *L*, scan length, is determined from *S/M*. Therefore, the size of the distance matrix is *L* × *L* not *S* × *S*. The computation complexity depends not on the number of scan cells but on the scan length, which is the number of scan cells in a scan chain. Consequently, the computational time of the proposed method is not drastically increased as the number of scan cells is increased for large designs. The experiments were conducted to investigate how the computational time varies in proportion to the number of scan chains. Table 5 presents that the computational time was decreased as the number of scan chains increased without affecting the power reduction effect. For the or1200 circuit with 10 scan chains, the computational time was dramatically reduced by seven times. However, for [28], scan chain reordering is performed for all scan cells, and scan cells are assigned to each scan chain. Therefore, the computational time is constant regardless of the number of scan chains. Therefore, the proposed method exhibits superior computational time compared with the previous method.

### 5.5. Test Compression Mode

In the experiments, the power reduction effect and test coverage of the proposed method were measured when using the test compression architecture. The experiments were conducted to compare the conventional test compression with the test compression of the proposed method for four ISCAS’89 and one OpenCores benchmark circuits. The experiments were conducted with the compression rate set to 60%. As shown in Table 6, the power consumption was decreased by 7.57% on average in the benchmark circuits compared with the conventional test compression. In these experimental results, it can be seen that the power reduction effect is relatively low compared to when the test compression is not applied. This is because the transition of shift-in operation cannot be reduced due to the pseudo-random patterns of the test compression and newly created test patterns are used with the proposed method in the test compression. However, the proposed method still can reduce the power consumption during the scan testing and does not change the test coverage significantly. Therefore, it was confirmed that the proposed method with the test compression also has the power reduction effect and does not affect the test coverage.

### 5.6. Performance Overhead

The proposed method can achieve a significant power reduction effect during scan testing by reordering the highly correlated scan cells to be adjacent, and additional hardware is not required because only the order of scan cells is changed. Moreover, the same test patterns can be used to test the circuits. Therefore, the test coverage of the proposed method is equivalent to that of [26,27,28] and the conventional stitching method. The test times also remain the same during scan testing. However, when changing the order of scan cells significantly, the routing length increases compared with that of the conventional stitching method based on the physical design—resulting in performance overhead (e.g., latency). For example, after the order of the scan cells is determined, when stitching the scan chain according to the order, if the location between the scan cell and the scan cell is physically far apart, requiring a long wire length and making routing inefficient. This performance overhead can be important to the design specifications and hold/setup time violation for flip-flop. Therefore, it is important to check how much the routing length increases when the proposed method is applied. The experiments were conducted to measure the routing length of the proposed method.

Table 7 presents the comparison of the routing length for four ISCAS’89 circuits and one OpenCores circuit. The experimental results show the comparison of the routing length of the conventional stitching and the proposed method, using the number of scan partitions which is proper by considering the number of flip-flops in the benchmark circuits. As shown in Table 7, except for s38584 benchmark circuit, it is confirmed that the proposed method has the routing overhead of 15.93% on average in s13207, s15850, s38417, and or1200 benchmark circuits. Although the s38584 circuit has the routing overhead of about 30%, considering that the impact on the routing length is small in or1200 benchmark circuit, which is a relatively large circuit, the impact on the routing length by the proposed method can be seen to be small. Additionally, the shift clock cycles are longer than the functional clock cycles, the latency generated by the routing overhead does not significantly impact the scan speed because shift clock cycles are sufficiently long. Consequently, the performance overhead is acceptable when considering the size of the routing overhead and the power reduction effect of the proposed method.

## 6. Conclusions

In this paper, a novel scan correlation-aware scan cluster reordering method is proposed for low test power in cryptographic circuits for wireless sensor networks. The novelty of the proposed method is that it performs scan correlation clustering to place high-correlation scan cells adjacent to each other. The algorithm for the proposed method includes three steps. First, a pairwise distance matrix between scan cells is constructed using test patterns. Second, scan correlation clustering is performed using the distance matrix, and a dendrogram tree is obtained as a clustering result. Third, scan chain reordering is performed by traversing the leaves of the dendrogram tree. The proposed method is expected to achieve, on average, an approximate 70% reduction in test power compared with the conventional method, reduce power consumption by 28.31%, and reduce computational time by 91.72%, for all experimental benchmark circuits compared with the previous methods. Consequently, through significant power reduction effect, the proposed method can improve the reliability of cryptography circuits and ensure the security and integrity of information in WSNs.

## Figures and Tables

**Figure 1 sensors-21-06111-f001:**
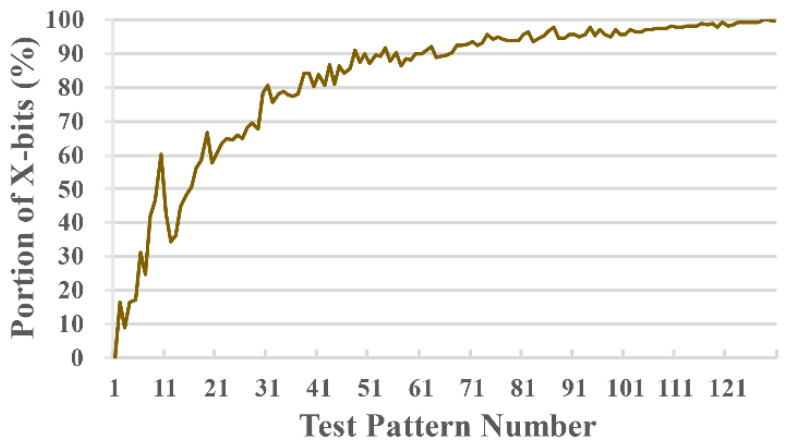
Portion of the X-bits in ISCAS benchmark s38584 [29].

**Figure 2 sensors-21-06111-f002:**
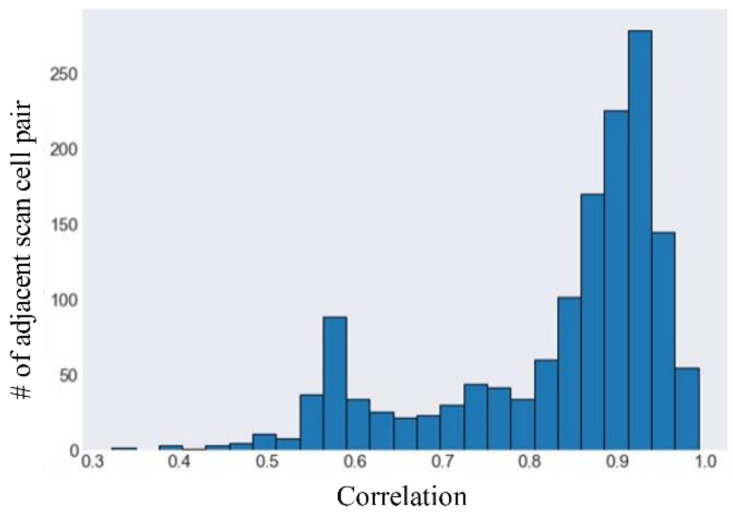
Correlation distribution of adjacent scan cells in s38584.

**Figure 3 sensors-21-06111-f003:**
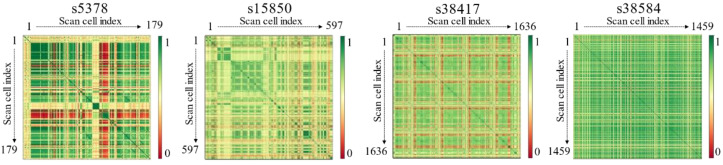
Heatmap graph of scan cell-to-cell correlation.

**Figure 4 sensors-21-06111-f004:**
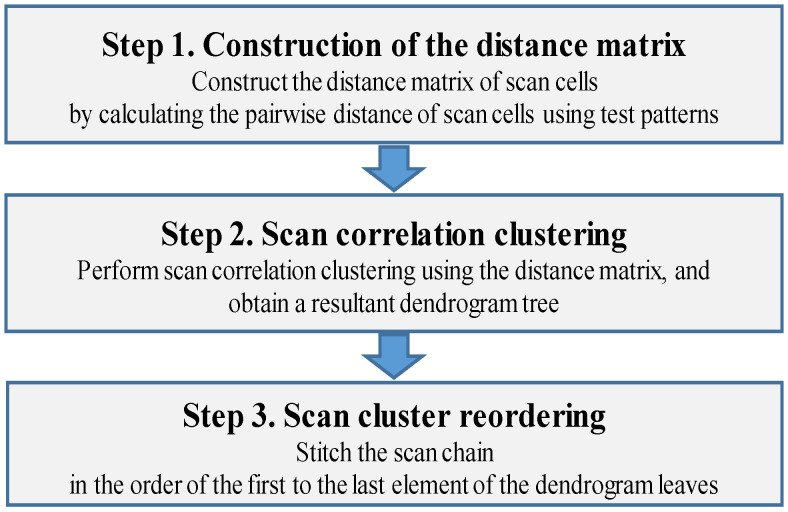
Overall flow of the proposed method.

**Figure 5 sensors-21-06111-f005:**
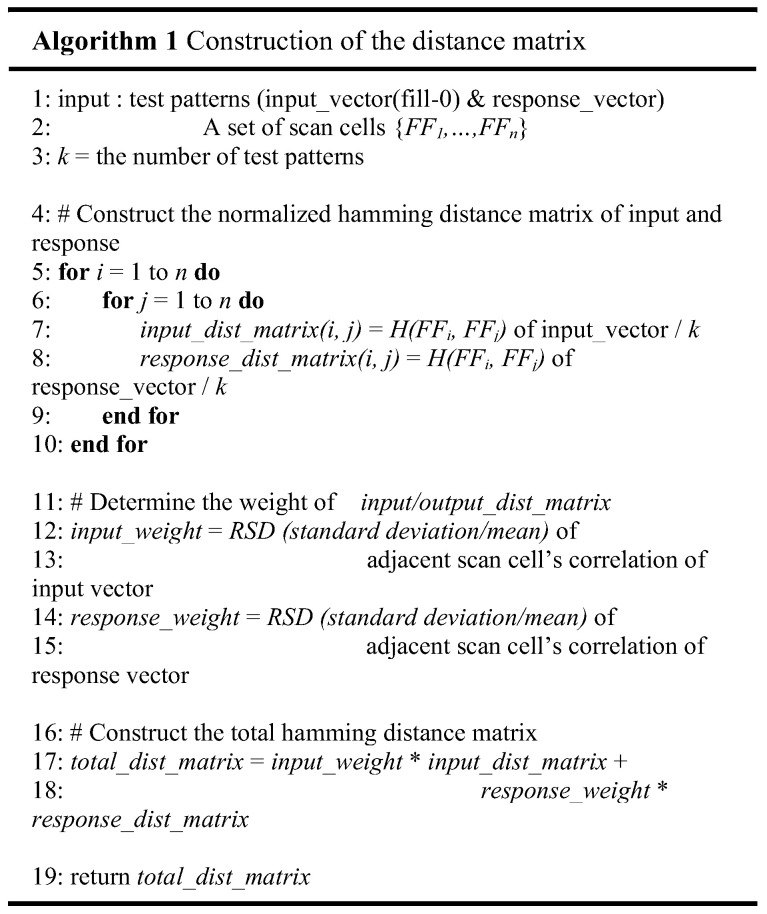
Algorithm for constructing the distance matrix of scan cells.

**Figure 6 sensors-21-06111-f006:**
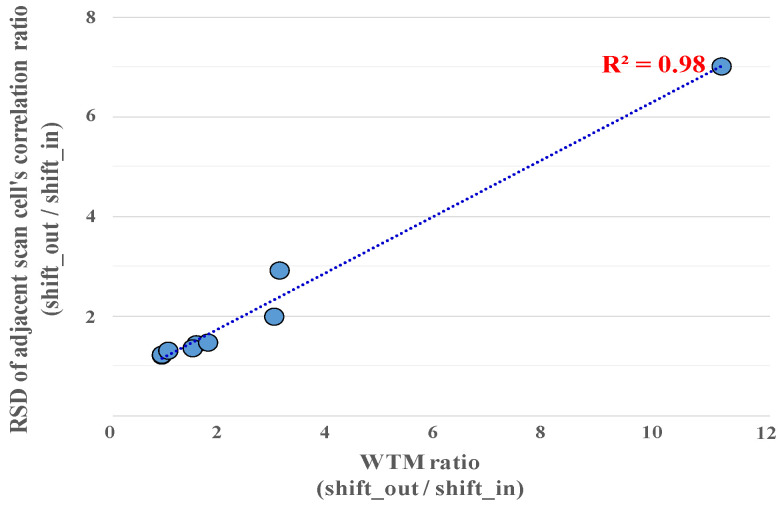
WTM and relative standard deviation (RSD) of adjacent scan correlation.

**Figure 7 sensors-21-06111-f007:**
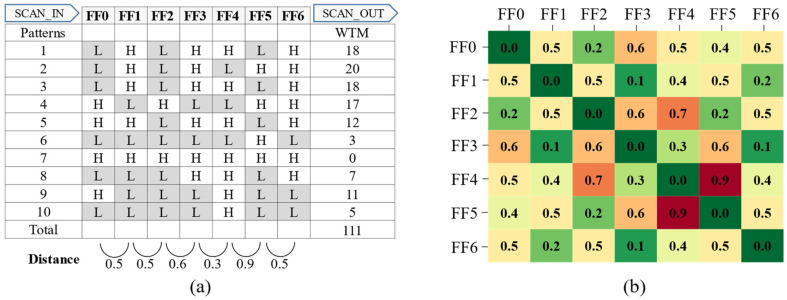
(**a**) WTM of example response vector with conventional stitching applied. (**b**) Distance matrix of scan cells in example response vector.

**Figure 8 sensors-21-06111-f008:**
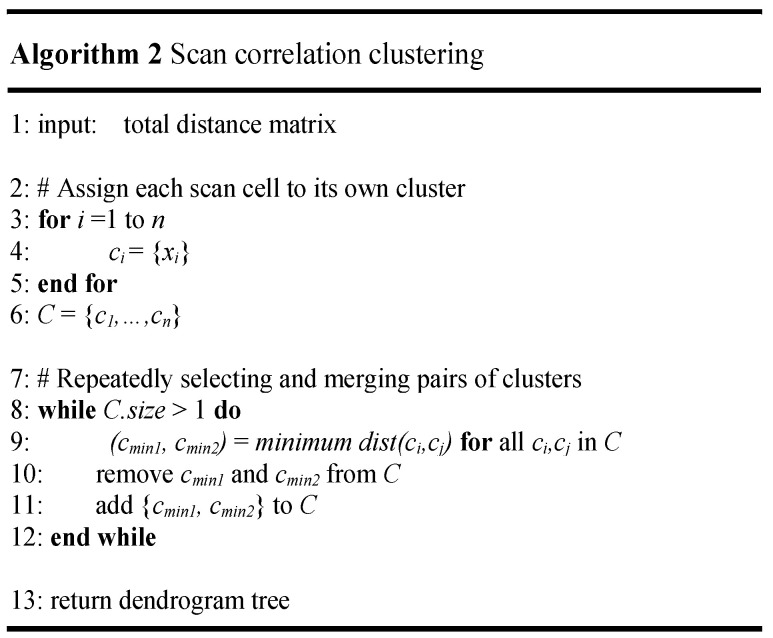
Algorithm for scan correlation clustering.

**Figure 9 sensors-21-06111-f009:**
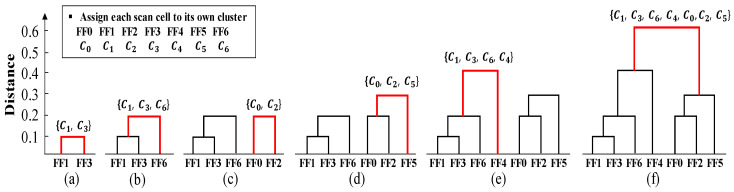
Scan correlation clustering process from (**a**) to (**f**) for dendrogram tree.

**Figure 10 sensors-21-06111-f010:**
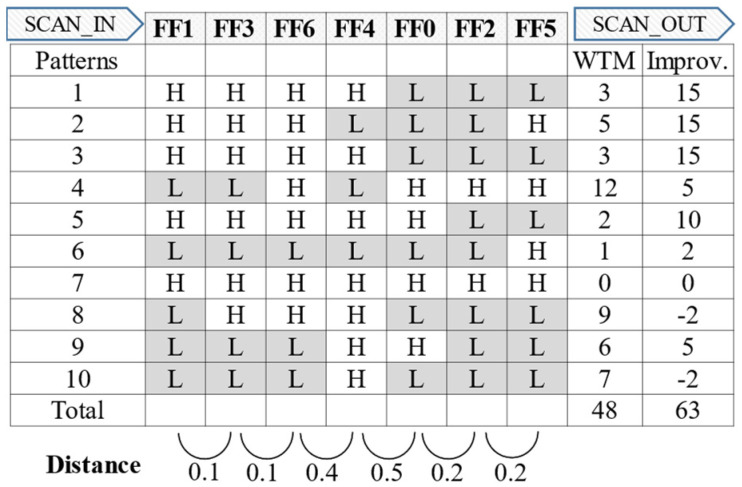
WTM of example response vector with the proposed method.

**Figure 11 sensors-21-06111-f011:**
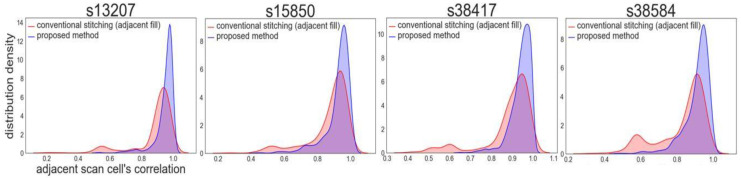
Comparison of the correlation distributions of the adjacent scan cells.

**Table 1 sensors-21-06111-t001:** Comparison of the correlations of adjacent scan cells and power consumption.

Benchmark	CUT	Number of Patterns	Conventional Stitching Method (Adjacent Fill)					Proposed Method						
			Correlation of Adjacent Scan Cells			WTM Value	Test Coverage	Correlation of Adjacent Scan Cells				WTM		Test Coverage
			Stdev	Mean	RSD (Stdev/Mean)			Stdev	Mean	RSD (Stdev/Mean)		Value	Reduction Rate	
										Value	Reduction Rate			
ISCAS’89	s13207	99	0.133	0.888	0.150	6,946,505	100.00%	0.049	0.952	0.051	65.63%	2,511,946	63.84%	100.00%
	s15850	102	0.125	0.872	0.143	4,558,619	99.99%	0.076	0.926	0.082	42.75%	2,572,503	43.57%	99.99%
	s38417	339	0.117	0.901	0.130	84,642,514	100.00%	0.039	0.926	0.042	67.57%	30,839,991	63.56%	100.00%
	s38584	121	0.132	0.830	0.159	43,629,699	100.00%	0.070	0.911	0.077	51.68%	21,983,926	49.61%	100.00%
OpenCores	or1200	1948	0.057	0.986	0.058	445,492,054	99.67%	0.015	0.995	0.015	73.92%	130,072,029	70.80%	99.67%
	Average reduction rate										60.31%		58.28%	

**Table 2 sensors-21-06111-t002:** Comparison of power consumption with the previous method.

CUT	Number of Scan Cells	Number of Patterns	Previous Method				Proposed Method		
			[26]	[27]	[28]	WTM	Reduction Rate Over [26]	Reduction Rate Over [27]	Reduction Rate Over [28]
s13207	669	99	3,663,689	2,867,436	6,678,015	2,511,946	31.44%	12.40%	62.38%
s15850	597	102	2,974,077	3,317,747	4,487,265	2,572,503	13.50%	22.46%	42.67%
s38417	1636	339	39,275,921	34,234,908	38,107,157	30,839,991	21.48%	9.92%	19.07%
s38584	1452	121	37,449,982	25,360,926	43,717,854	21,983,926	41.30%	13.32%	49.71%
	Average reduction rate						26.93%	14.53%	43.46%

**Table 3 sensors-21-06111-t003:** Power reduction effect according to x-bit portion.

Benchmark	CUT	Number of Scan Cells	X-bit Portion in Input Vector	WTM					
				Conventional Stitching Method (0 Fill)		Conventional Stitching Method (Adjacent Fill)		Proposed Method	
				Scan-In	Scan-Out	Scan-In	Scan-Out	Scan-In	Scan-Out
ISCAS’89	s13207	669	91%	1,422,076	4,132,997	1,230,024	5,716,481	1,272,636	1,239,310
	s15850	597	80%	1,577,606	3,102,140	1,244,078	3,314,541	1,207,100	1,365,403
	s38417	1636	93%	20,182,921	59,813,361	13,321,286	71,312,228	15,814,621	15,025,370
	s38584	1452	82%	14,614,319	29,017,581	11,576,776	32,052,923	10,043,223	11,940,703
OpenCores	or1200	3430	99%	51,996,870	364,592,450	44,806,688	400,685,366	100,638,047	32,582,709

**Table 4 sensors-21-06111-t004:** Comparison of computational time with the previous method.

Benchmark	CUT	Number of Scan Cells	Computational time (s)		
			[28]	Proposed Method	Reduction Rate
ISCAS’89	s13207	669	10.5	1.8	82.86%
	s15850	597	16.8	1.7	89.88%
	s38417	1636	133.3	8.5	93.62%
	s38584	1452	120.5	6.2	94.85%
OpenCores	or1200	3430	2692.3	70.3	97.39%
	Average reduction rate				91.72%

**Table 5 sensors-21-06111-t005:** Comparison of computational time with the previous method.

Benchmark	CUT	Number of Scan Cells	Number of Scan Chains	WTM			Computational Time (s)
				Conventional Stitching Method (Adjacent Fill)	Proposed Method	Reduction Rate	
ISCAS’89	s13207	669	1	6,946,505	2,511,856	63.84%	1.8
			3	6,851,174	2,486,976	63.70%	1.5
			5	6,742,775	2,554,837	62.11%	1.5
			10	7,059,290	2,572,405	63.56%	1.0
	s15850	597	1	4,558,619	2,572,503	43.57%	1.7
			3	4,071,616	2,348,508	42.32%	1.2
			5	4,595,110	2,627,943	42.81%	1.2
			10	4,694,334	2,603,947	44.53%	1.0
	s38417	1636	1	84,642,514	30,839,991	63.56%	8.5
			3	83,243,769	30,933,385	62.84%	4.8
			5	83,638,924	30,912,946	63.04%	3.5
			10	84,079,332	30,167,664	64.12%	3.3
	s38584	1452	1	43,629,699	21,983,926	49.61%	6.2
			3	42,084,699	21,450,571	49.03%	3.5
			5	41,195,417	21,079,695	48.83%	2.6
			10	41,604,006	21,251,326	48.92%	2.4
OpenCores	or1200	3430	1	445,492,054	130,072,029	70.80%	70.3
			3	436,582,213	133,026,600	69.53%	21.4
			5	423,217,451	125,695,583	70.30%	16.2
			10	420,989,991	120,613,632	71.35%	9.4

**Table 6 sensors-21-06111-t006:** Comparison of power consumption with the test compression.

CUT	Conventional Method			Proposed Method			
	WTM	Number of Patterns	Test Coverage	WTM	Reduction Rate	# of Patterns	Test Coverage
s13207	29,240,413	234	98.25%	26,745,037	8.53%	201	98.30%
s15850	33,484,768	276	99.20%	31.392.612	6.24%	279	98.94%
s38417	332,923,984	440	99.37%	304,752,793	8.46%	366	99.37%
s38584	241,802,651	402	99.49%	222,196,267	8.10%	335	99.50%
or1200	1,283,287,056	967	99.56%	1,199,455,085	6.53%	960	99.56%

**Table 7 sensors-21-06111-t007:** Comparison of the routing length.

CUT	Conventional Method		Proposed Method	
	$\mathrm{Area}\;(um^{2})$	Net Length (um)	$\mathrm{Area}\;(um^{2})$	Net Length (um)
s13207	1257.85	14,025.47	1261.98	15,340.39
s15850	1260.02	13,874.40	1330.57	16,980.72
s38417	3534.63	40,089.63	3485.48	45,237.35
s38584	3752.28	47,762.39	3774.31	64,565.51
or1200	11581.29	234,236.28	12,540.06	279,040.38

## Data Availability

Not applicable.

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
