# Peer review of "Low-Power Scan Correlation-Aware Scan Cluster Reordering for Wireless Sensor Networks"

_sensors, 2021, doi:10.3390/s21186111_

Round 1

Reviewer 1 Report

Below I am presenting my comments to the authors:

1/ In the Abstract, be more precise on defining "Although many techniques have been proposed ...".

2/ Also in the Abstract, present your contributions and additions to the life of the people.

3/ Your introduction is nice, but it is overcharged, with much content, so being sometimes dense. I want to suggest you be more direct, detailing in only 1 paragraph the initiatives in the literature and their problems. Moreover, you could have a section 3 named Related work to describe your concurrents deeply.

4/ Again, present in the introduction the relevance of your research. Where it changes people's lives?

5/ In your proposed model, I suggest including a section named Design Decisions, with the premises of the model. Maybe combining this with the Overall flow.

6/ In the Overall flow, present the user's difficulty in inputting data to the system in detail? What is his/her level of expertise? What is automatic and transparent and what is not?

7/  How is the scalability and fault tolerance of your proposal?

8/ How are you dealing with radio frequency interoperation from other technologies and ranges?

9/ What is the time complexity of your solution. In other words, what are the time to compute it and the periodicity of this trigger?

10/  It is essential to present a section named Evaluation Methodology, with:
Prototype/Simulation
Scenarios - comparisons, parameters
Metrics
Infrastructure

11/ What are the main limitations of your proposal? What are the paths of not following, for example?

12/ In the Conclusion, present the contributions to society. Also, May I use your ideas on other areas beyond WSN?

13/ Critical: You are outdated. How about references from 2021 and 2020. I could reject your article! Again, present a related work section with updated articles. 

Author Response

we have upload answers as a file.

Reviewer 2 Report

In this paper, a novel scan correlation- aware scan cluster reordering is proposed to solve this problem. 
The study is interesting and the selected papers are good. 
However, the paper needs to acquire more quality in terms of cited papers. 
For this purpose I suggest to include the following paper among the cited papers because, in my opinion, it is crucial introducing some of dependability problems of wireless sensor networks (WSNs):

"Avr-inject: A tool for injecting faults in wireless sensor nodes" 2009. IEEE International Symposium on Parallel & Distributed Processing, IPDPS 2009.  

I am confident if the authors add the citation and they motivate the importance to consider it, the paper will acquire more quality for the publication. 

Author Response

we have uploaded answers as a file. 

Round 2

Reviewer 1 Report

The authors have addressed all my previous issues. In special, now we have the motivation and contribution presented in a clearer way in both Abstract and Introduction. Also, I perceived that we hae improvements in the Abstract, highlighting the gaps in the area. One of the biggest modifications was Subsection 4.2, with the big picture of the system. Evaluation Methodology was improved, together with the description of the results. That said, I am confident that we have a good work to be published, with a good quality and contribution to readers.